# Outcomes and Quality of Life of Systemic Therapy in Advanced Hepatocellular Carcinoma

**DOI:** 10.3390/cancers11060861

**Published:** 2019-06-21

**Authors:** Kehua Zhou, Christos Fountzilas

**Affiliations:** 1Catholic Health System Internal Medicine Training Program, Jacobs School of Medicine and Biomedical Sciences, University at Buffalo, Buffalo, NY 14214, USA; kehuazho@buffalo.edu; 2Division of Gastrointestinal Medicine, Department of Medicine, Roswell Park Comprehensive Cancer Center, Buffalo, NY 14263, USA

**Keywords:** hepatocellular carcinoma, multikinase inhibitors, immunotherapy, quality of life, drug toxicity

## Abstract

Hepatocellular carcinoma (HCC) is one of the most commonly diagnosed cancers worldwide; most patients are diagnosed with advanced disease for which there is no known cure. Tremendous progress has been made over the past decade in the development of new agents for HCC, including small-molecule kinase inhibitors such as sorafenib, lenvatinib, cabozantinib, regorafenib, and monoclonal antibodies like ramucirumab, nivolumab, and pembrolizumab. Ideal use of these agents in clinics has improved the long-term outcome of patients with advanced HCC as well as introduced unique toxicities that can affect quality of life. These toxicities usually are thought to be partially related to cirrhosis, a major risk factor for the development of HCC and a pathophysiological barrier complicating the optimal delivery of antineoplastic therapy. Additionally, side effects of medications together with advanced HCC symptoms not only decrease quality of life, but also cause treatment interruptions and dose reductions that can potentially decrease efficacy. Physicians caring for patients with advanced HCC are called to optimally manage HCC along with cirrhosis in order to prolong life while at the same time preserve the quality of life. In this review, we aimed to summarize outcomes and quality of life with the use of modern systemic treatments in advanced HCC and provide a physician reference for treatment toxicity and cirrhosis management.

## 1. Introduction

Hepatocellular carcinoma (HCC) is one of the leading causes of cancer-related death worldwide [1]. It is the most common cancer in Northern and Western Africa and Eastern and South-Eastern Asia; however, its incidence has been recently increasing in the United States with an estimated 42,030 new cases and 31,780 HCC-related deaths in 2019 [2,3]. Risk factors include hepatitis B virus (HBV) or hepatitis C virus (HCV), aflatoxin-contaminated foods, heavy alcohol intake, obesity, smoking, and type 2 diabetes. Chronic inflammation causes cirrhotic and non-cirrhotic changes in the liver and eventually leads to HCC. While the majority of HCC arises from cirrhosis, 20% of HCC involves a noncirrhotic liver [4]. Cirrhosis is the result of inflammation, necrosis, fibrosis, and ongoing regeneration of hepatocytes in the liver; non-cirrhotic HCC involves alterations in cell cycle regulation, oxidative stress, and increased levels of tumorigenic growth factors [5]. 

Unfortunately, the majority of patients with HCC are diagnosed late in its course, rendering them unsuitable for curative therapy. Over the past decade, tremendous development has been made in drug development for advanced HCC [6]. The oral small molecule multikinase inhibitors (MTKis) sorafenib, cabozantinib, regorafenib, lenvatinib, and the monoclonal antibody (mAb) ramucirumab have proven efficacy as first- or second-line therapies in phase III clinical trials [7,8,9,10,11,12], and were approved by the United States Food and Drug Administration (FDA) for advanced HCC [13,14,15,16,17]. Meanwhile, immunotherapy has shown promising results [18,19], and the FDA conditionally approved nivolumab [20] and pembrolizumab [21] in the second-line setting. While the overall survival (OS) for patients with advanced HCC has increased, these treatments are not curative [3] and the unique treatment-related toxicities can further compromise patients’ delicate health. As such, treatment strategies maintaining adequate quality of life (QoL) are highly valued. QoL refers to the state of complete physical, mental, and social wellbeing, and not merely the absence of disease or infirmity. Patient-reported baseline QoL was found to be a prognostic marker for survival for patients with unresectable HCC [22]. Symptoms affecting QoL like appetite loss, decreased physical function and role function, and fatigue have all been found to correlate with shorter OS [23]. As a result, an increasing number of researchers advocate using health related QoL as an independent prognostic factor for response to treatment and progression of disease in advanced HCC [24,25]. 

In this review article, we aimed to summarize outcomes and QoL with the use of modern, life-prolonging systemic treatments in advanced HCC and provide a physician reference for treatment toxicity and cirrhosis management.

## 2. Methods

Using the search logic, ("quality of life") AND hepatocellular carcinoma [Title/Abstract], we identified 667 articles in PubMed. Abstracts were then screened to identify outcome and QoL results. The search for systemic medications was guided by the FDA approval information [13,14,15,16,17,20,21]. Specifically, we searched for and identified all phase III trials in PubMed for targeted therapy for HCC with proven OS benefit (including sorafenib, cabozantinib, lenvatinib, regorafenib, and ramucirumab [7,8,9,10,11,12]). Using the search logic, ((immune therapy) OR immunotherapy) AND hepatocellular carcinoma [Title] with article type limited to clinical trials, we were able to locate 80 articles with immune checkpoint inhibitors (ICIs) in PubMed, and we were only able to identify and include two single-arm phase II trials [18,19]. Finally, we searched Google and Google Scholar for the same information; we were able to identify and include one additional phase III trial testing an FDA approved agent (KeyNote-240) [26]. The search was carried out from inception to June 6, 2019. Data on outcomes, QoL, and toxicities in advanced HCC of these trials were extracted in Table 1. Additionally, we also searched for cost effectiveness analyses of these medications and relevant information, if any, was described and presented separately in Table 2.

## 3. Oral Multikinase Inhibitors

### 3.1. Sorafenib

Sorafenib inhibits vascular endothelial growth factor receptor (VEGFR)1-3, platelet-derived growth factor receptor (PDGFR), KIT, and RET and downstreams Raf signaling molecules of Raf-1 and B-Raf [32]. Sorafenib has been approved as the first-line treatment for advanced HCC since 2007 [33] based on the landmark Sorafenib Hepatocellular Carcinoma Assessment Randomized Protocol (SHARP) trial [7] and the follow-up Sorafenib Asia-Pacific trial with proven longer OS over placebo [8]. However, patient-reported outcomes—which may better reflect the impact of treatment-related toxicity and response to tumor-related symptoms—did not differ between sorafenib and placebo groups [7,8]. Notable side effects of sorafenib include hand-foot skin reaction (HFSR) (7.1%), diarrhea (18.5%), anorexia (77.1%), fatigue (15.5%), rash or desquamation (6.4%), weight loss (10.1%), and hypertension (3.0%) [34]. 

In order to maximize benefits while minimizing adverse events, a panel of Italian experts convened in April 2013 and recommended the following: (1) sorafenib should be initiated at the approved dose of 400 mg twice a day; (2) discontinuation of sorafenib should be based on assessment of patients’ overall response including symptomatic progressions rather than radiologic response alone [35]. However, it is worth noting that in a previous study, Iavarone et al. [36] found that the median survival was 21.6 months with a half dose versus 9.6 months with a full dose of sorafenib; and in a follow-up study, Ponziani et al. [37] found that sorafenib dose adjustments based on tolerability increased sorafenib exposure and optimized outcomes. More recently, in a retrospective study, Reiss et al. [38] reported that reduced starting dose (<400 mg twice daily) of sorafenib as compared to standard dosing resulted in reduced pill burden and treatment costs with a tendency for longer OS.

As a first-line agent for advanced HCC for over 10 years, multiple cost-effectiveness analyses have been performed for the use of sorafenib in advanced HCC [27]. Motevailli et al. [27] summarized these cost-effectiveness analyses and found that the use of sorafenib resulted in an incremental quality adjusted life year (QALY) of 0.18 to 0.53 as compared with best supportive care with incremental costs per QALY in the range of –1,014,507 to 934,803 (USD).

### 3.2. Lenvatinib

Lenvatinib inhibits fibroblast growth factor (FGF) receptors (FGFR1–4), VEGFR 1–3, PDGFRα), KIT, and RET [39,40]. Its use as a first-line treatment in advanced HCC was approved by the FDA on August 16, 2018 [16] after proving to be non-inferior to sorafenib in terms of OS [9]. Additionally, lenvatinib showed statistically significant improvements compared with sorafenib in some QoL assessments such as the time to clinically meaningful deterioration in role functioning, pain, diarrhea, nutrition, and body image, although the summary score of QoL measurements was not significantly different between sorafenib and lenvatinib (hazard ratio (HR) 0.87, 95% CI 0.754–1.013, *p* = 0.07). Common side effects of lenvatinib included hypertension (42%), diarrhea (39%), decreased appetite (34%), and decreased weight (31%) [9]. Kobayashi et al. [28] compared lenvatinib with sorafenib and found that the use of lenvatinib led to a gain of 0.27 incremental life year and 0.23 QALY improvement and lowered the costs by 406,307 Japanese Yen (3,659.2 USD).

### 3.3. Regorafenib

Regorafenib is an oral inhibitor of VEGFR1-3, tyrosine kinase with immunoglobulin-like and epidermal growth factor receptor-like domains 2 (TIE2), PDGFRβ, FGFR, KIT, RET, RAF-1, and BRAF [41,42]. On April 27, 2017, the FDA approved regorafenib as a second-line treatment for advanced HCC [15]. For patients with HCC who progressed on sorafenib, regorafenib improved OS and progression-free survival (PFS) compared to placebo (the RESORCE trial) [10]. Common side effects of regorafenib include hypertension (15%), HFSR (13%), fatigue (9%), and diarrhea (3%) [10]. No difference was found between regorafenib and placebo for QoL measures with the exception of the Functional Assessment of Cancer Therapy-Hepatobiliary Questionnaire total score favoring the placebo (regorafenib vs. placebo = 129.31 vs. 133.17, *p* < 0.001, lower score indicating worse QoL) [43]. It has to be noted that the original trial excluded patients who were intolerant to side effects of sorafenib [10]. Based on these data, Shlomai et al. [29] predicted that the use of regorafenib can lead to a gain of 19.76 weeks of life (0.38 life years) and 0.25 QALYs; and Parikh et al. [30] reported that regorafenib provided an increase of 0.18 QALYs at a cost of $47,112 (USD) for patients with advanced HCC. 

### 3.4. Cabozantinib

Cabozantinib inhibits VEGFR 2, c-MET, RET, c-KIT, AXL, TIE2, and FLT3. On January 14, 2019, the FDA approved cabozantinib as a second-line treatment in patients with advanced HCC [14]. In patients with advanced HCC who had disease progression after one to two systemic treatments, cabozantinib resulted in longer OS and PFS, and a slightly higher objective response rate compared to placebo [11]. Common side effects associated with cabozantinib include HFSR (17%), hypertension (16%), increased aspartate aminotransferase (AST) level (12%), fatigue (10%), and diarrhea (10%) [11]. QoL results were not originally reported for this trial. In a follow-up abstract publication, Abou-Alfa et al. [31] reported that QALYs were slightly reduced at day 50 favoring the placebo, but increased to +0.007 QALYs at day 100 and +0.032 QALYs at day 150 and + 0.092 additional QALYs over the entire follow-up favoring cabozantinib. These results were based on results of EQ-5D, a measure of health-related QoL, which consists of a descriptive system and the EuroQol-visual analogue scales.

## 4. Monoclonal Antibodies 

### 4.1. Ramucirumab 

Ramucirumab is a fully human IgG1 mAb that inhibits tumor growth and metastases via blocking ligand activation of VEGFR2 [44]. The results of the first positive phase III clinical trial of ramucirumab for advanced HCC (REACH-2) were published in February 2019 [12]. Based on results of this trial, ramucirumab was approved in May 2019 by the FDA as a second-line agent for advanced HCC (after treatment with sorafenib) with alpha-fetoprotein (AFP) of ≥400 ng/mL [17]. No between-group difference was found in the deterioration of symptoms and these performance status measures. However, in a follow-up secondary analysis, ramucirumab significantly reduced deterioration in the Functional Assessment of Cancer Therapy Hepatobiliary Symptom Index 8 (FHSI-8) at the end of treatment compared with placebo (*p* = 0.0381) with a trend favoring a delay in the deterioration of symptoms (HR 0.690; *p* = 0.054) and performance status (HR 0.642; *p* = 0.057) [45]. Common grade 3 or worse side effects included hypertension (13%), hyponatremia (6%), and increased AST (3%) [12]. 

### 4.2. Immunotherapy

Nivolumab and pembrolizumab are the two most widely studied human IgG4 mAbs against programmed cell death protein 1 (PD-1) [46]. They have been both approved by the FDA as second-line therapies for advanced HCC under the provisions of accelerated approval which indicates further studies are required for their use in advanced HCC refractory to sorafenib [20,21]. The approval of nivolumab and pembrolizumab for HCC by the FDA was based on early reports of phase I/II trials indicating a promising response in patients with HCC (Table 1) [18,19]. 

In the CheckMate 040 trial [18], nivolumab showed a manageable safety profile with acceptable tolerability. Common side effects of nivolumab included fatigue, diarrhea, pruritus, rash and liver function test abnormalities. Serious side effects of pemphigoid, adrenal insufficiency, and liver disorder were also reported in cases. QoL was assessed with multiple questionnaires and demonstrated consistently stable findings with no significant change from the baseline. The phase III CheckMate 459 study evaluating nivolumab in the first-line setting vs. sorafenib is ongoing. Efficacy and safety of pembrolizumab were assessed in the KeyNote-224 trial where 104 patients with HCC previously treated with sorafenib were enrolled [19]. Toxicities were manageable. Apart from side effects described in Table 1, cases of adrenal insufficiency, elevated bilirubin concentration, cholestatic jaundice, and increased alanine aminotransferase were also reported. In KeyNote-040 and KeyNote-224 studies, both agents showed promising antineoplastic activity with objective response ranging between 15%–20% and median OS of approximately a year. The preliminary results of the phase III study (KeyNote-240) of pembrolizumab versus best supportive care for second-line therapy in advanced HCC were presented in the 2019 ASCO Annual Meeting [26]; pembrolizumab improved OS (HR: 0.78; one sided *p* = 0.0238) and PFS (HR: 0.78; one sided *p* = 0.0209) with a manageable toxicity profile compatible with prior experience with ICIs in advanced HCC; however, these differences did not meet significance per the prespecified statistical plan. QoL information was reported in neither Keynote-224 nor Keynote-240. 

## 5. The Role of Cirrhosis

Cirrhosis affects approximately 0.27% of the population, corresponding to 633,323 adults in the United States (based on 2010 US census data). Compared to matched controls, patients with cirrhosis has an 18% higher risk of mortality per 2-year interval [47]. A cirrhotic liver reduces effective blood flow through intrahepatic shunts and sinusoidal capillarization. It decreases CYP enzyme secretions, alters enzyme inductions, and impairs the clearance of drugs [48]. Approximately 80% of patients with HCC have underlying cirrhosis [4]. Consequently, systemic treatments of HCC will often inevitably involve considerations and treatment of cirrhosis. 

Treatment of compensated cirrhosis usually aims at the underlying liver disease including treatment of viral hepatitis, alcohol abstinence, and avoidance of hepatotoxic drugs in addition to symptom control. In decompensated cirrhosis, the removal of etiological factors, particularly alcohol consumption and hepatitis B or C virus infection, is associated with decreased risk of decompensation and increased OS [49]. It requires additional complication-specific interventions, for example, nonselective beta-blocker and endoscopic band ligation for esophageal varices; vasoactive drugs (terlipressin, somatostatin or octreotide) together with endoscopic variceal ligation or sclerotherapy for variceal bleeding; diuretics (spironolactone alone or combination with furosemide or torsemide), and sodium restriction (80–120 mmol/day, corresponding to 4.6–6.9 g of salt) for ascites; terlipressin in combination with albumin (non-intensive care unit) or norepinephrine (intensive care unit) and less favorably midodrine, octreotide, and albumin for hepatorenal syndrome; antibiotic treatment and prophylaxis for spontaneous bacterial peritonitis; albumin or baclofen administration (10 to 30 mg/day) for muscle cramps; and transjugular intrahepatic portosystemic shunts (TIPS) for portal hypertension and refractory variceal bleeding [49]. TIPS is associated with the occurrence of shunt stenosis and hepatic encephalopathy. The management of hepatic encephalopathy targets plasma ammonia primarily by reducing its production and absorption. Lactulose and rifaximin have remained the mainstay of treatment [50]. Additionally, repeated large volume paracentesis plus albumin (8 g/L of ascites removed) may be required for refractory ascites, and the use of nonselective beta-blockers should be cautious in severe or refractory ascites [49].

Morbidity and mortality in patients with HCC are related to both the malignancy and the presence of chronic liver disease. Prophylaxis and treatment of decompensated cirrhosis has been shown to decrease HCC-specific mortality and morbidity in many studies [51,52,53,54,55,56,57]. Nonetheless, in a recent study, Cabibbo et al. [58] argued that treatment of hepatitis C in patients with "active" HCC remains controversial as these patients were usually excluded in clinical trials. For HBV-related HCCs, treatment of hepatitis can preserve or improve liver function, and may be beneficial for patients undergoing curative therapy, or locoregional therapy/chemotherapy with reasonable life expectancy [59]. In HCC patients with variceal bleeding, 1-year and 2.5-year survival rates were 56.6% and 28.3%, respectively [52]. Kim et al. [60] found that primary prophylaxis using a non-selective beta blocker or upper endoscopy with ligation or sclerotherapy for variceal bleeding was associated with a reduced risk of mortality in all patients with HCC (HR 0.54; 95% CI 0.33–0.88; *p* = 0.014). Meanwhile, endoscopic interventions combined with pharmacological therapy as compared with pharmacological therapy alone was found to improve the OS in HCC patients with esophageal variceal bleeding [57]. 

## 6. Management of Treatment Toxicity and Other Supportive Care 

Literature on drug effect optimization and drug side-effect management specifically related to the use of cabozantinib, regorafenib, lenvatinib, and ramucirumab, with the exception of sorafenib, in patients with advanced HCC remain scarce. Nonetheless, treatment of side effects of these medications is similar to those of sorafenib, which is mainly symptomatic management (Table 3). For severe adverse events or refractory side effects, dose adjustment or even drug discontinuation may be appropriate, and re-escalation to full dose when possible is recommended [35,61]. 

For the management of rash and HFSR, primary prevention is recommended to start early which includes callus removal and minimizing friction and direct trauma [61]. As for secondary prevention, keratolytic creams (containing urea, α-hydroxy acids or salicylic acid) can be used to aid natural exfoliation and emollient creams for moisturization on top of appropriate use of topical corticosteroids (clobetasol 0.05% ointment, cortisone cream) and analgesics (e.g., lidocaine 2%) and systematic strategies (e.g., pyridoxine 50–150 mg/day to reduce symptoms). For diarrhea management, loperamide of 4 mg followed by 2 mg every 2 h until 2 h after the last bowel movement has been recommended (on top of lactulose dose adjustment); other medications, like opiates (e.g., codeine) and cholestyramine 4 g three times a day have also been suggested. 

Pharmacological and non-pharmacological interventions including psychostimulants (methylphenidate and modafinil), treatment for pain, emotional distress, anemia and hypothyroidism, physical exercise, psychoeducation, cognitive behavioral therapy, other mind-body therapies, and activity enhancement may be appropriate on specific occasions for patients with fatigue. Hypertension in general should be managed per standard medical practice. Due to the added benefits of decreasing portal hypertension, special considerations should be given to non-selective beta blockers (e.g., carvedilol, nadolol or propranolol) followed by calcium channel blockers if non-effective. Interestingly, based on preclinical studies and small controlled trials, renin-angiotensin system inhibitors may have antitumor effects (likely related to VEGF response) and improve both liver fibrosis and portal hypertension; their use correlated with a gain of 5 months in median OS in patients with advanced HCC [62].

Pain, especially right upper quadrant abdominal pain, is common in patients with advanced HCC, and its management can be challenging as cirrhosis can alter drug pharmacokinetics. Pain management in patients with advanced HCC could be more liberating in opioid use (as compared to chronic non-cancer pain) [63]. Meanwhile, pain in these patients remains undertreated for fears of increased risks of toxicity [64]. Morphine, together with its derivatives, are the first-line choice with fentanyl as an excellent option in patients with impaired liver functions, and oxycodone/naloxone can be a safe and effective alternative [65]. Nonetheless, management of pain in HCC should follow a similar logical sequence as in chronic non-cancer pain, including balancing benefits and side effects of treatment options and progressing from using nonpharmacological to pharmacological and interventional pain management methods [66]. Steroids can reduce tumor-related edema, and have anti-inflammatory effects and direct effects on nociceptive neural systems; patients with cancer-related pain can be treated with 1–2 mg of dexamethasone (the alternative is prednisone) orally or parenterally twice daily, possibly preceded by a larger loading dose of 10–20 mg. Using steroids in patients treated with ICIs for reasons other than immune-related adverse events is discouraged as the potential effect on antitumor effect is unknown. Additionally, radiotherapy can also be used in patients with bone metastasis and to relieve symptoms from pulmonary or lymph node metastases [66]. 

Weight loss and malnutrition (may be related to both HCC itself and medication use) are almost universal in patients with advanced HCC. Early and structured multimodal interventions (including targeted nutritional supplementation, physical exercise, and pharmacological interventions) are usually appropriate but can be of little to no effect. In a phase III trial, Chow et al. [67] reported that megestrol acetate improved QoL, appetite, nausea/vomiting, and emotional functioning. Pharmacological agents including corticosteroids (dexamethasone, prednisolone, and methylprednisolone) and progesterone analogs (megestrol acetate and medroxyprogesterone acetate) have been used for weight loss and malnutrition due to cancer anorexia/cachexia in patients with advanced cancer [68]. Although parenteral nutrition, enteral nutrition or oral nutritional supplements lack compelling evidence support, they may be appropriate in highly selected patients who meet both of the following criteria: notable malnourishment or risk for notable malnourishment and potentially curable disease (apparently not advanced HCC) [69]. Again, steroid use in a patient on immunotherapy should be limited to the management of severe immune toxicity, where available data, although limited, did indicate no detrimental effect on the response to immunotherapy [70]. 

ICI therapy has similar systemic side effects, like fatigue and diarrhea, but also specific immune-mediated toxicities including dermatologic and mucosal toxicity, hepatotoxicity, pneumonitis, endocrinopathies, hypophysitis, adrenal insufficiency, and less common immune-related adverse events in other organs. Management of the common systemic side effects of ICI therapy is similar to those of MTKis as mentioned above; however, management of immune toxicity requires a different approach that emphasizes steroid use (Table 3). For immune-related toxicities affecting QoL and requiring intervention, ICI therapy is withheld, and prednisone 0.5 mg/kg/day or equivalent is recommended followed by slow taper upon resolution. In patients with severe or life-threatening immune-mediated toxicities, immune checkpoint inhibitor therapy should be permanently discontinued, and prednisone 1 to 2 mg/kg/day or equivalent should be given with a taper over at least one month after symptom relief [71]. Additionally, checkpoint inhibitor therapy should be withheld for AST or ALT >2.5 times the upper limit of normal (ULN) but ≤5 times the ULN, or total bilirubin >1.5 times the ULN but ≤3 times the ULN; and permanently discontinued for AST or ALT >5 times the ULN, or total bilirubin >3 times the ULN [71]. 

## 7. Discussion

Compared to the general population and patients with advanced chronic liver disease, patients with HCC have worse physical, functional, emotional, and social-family wellbeing, including poor energy, troubles with sleep, difficulties digesting food, loss of appetite, increased pain, declined physical mobility, decreased ability to perform usual activities, poor mental health and social functioning, and overall QoL [23,72]. Advanced cancer stage, older age, higher serum bilirubin levels, lower serum albumin levels, uncertainty of disease process, pain, fatigue, nausea, and poor performance status all have been identified as factors negatively affecting QoL [72]. 

After many decades of virtually no systemic options, we have now two first-line treatment options (sorafenib, lenvatinib), five second-line options (regorafenib, cabozantinib, ramucirumab, nivolumab, and pembrolizumab) and potentially one third-line option (cabozantinib); however, none is curative. How can we select the ideal treatment for our patients via balancing efficacy and toxicity so that QoL is preserved for a longer period of time? Differences in reported QoL metrics between studies are hard to interpret but lenvatinib resulted in clinically meaningful later deterioration than sorafenib when compared head to head [9], and ramucirumab delayed QoL deterioration compared to placebo [45]. Surprisingly, QoL continued to decline in patients enrolled in clinical trials of sorafenib and regorafenib (no difference with placebo), and in fact, symptomatic progression time seems shorter in sorafenib compared to placebo [7,8], and as compared to regorafenib, patients in the placebo group actually had better functions of daily life [10]. No QoL metrics using standard questionnaires are available for cabozantinib [11], but cost-effective analysis favors cabozantinib over the placebo [31]. Despite targeting the same, more, or less kinases for inhibition, MTKis can be quite different in terms of their side effect profile. For example, significant skin toxicity is more common with sorafenib compared to lenvatinib while the opposite is true for hypertension [9]. For patients with multiple cardiovascular risk factors, ICIs may be the preferred treatment option given their favorable adverse event profile (with the limitation that their relative effectiveness over MTKis is currently unknown), and MTKis with a low incidence of clinically significant hypertension may be used as well. 

Second, performance status and baseline liver function can significantly affect outcomes [73]. Approved MTKis and mAbs were evaluated in a highly selected population with Child–Pugh Class A (score 5 or 6) and Eastern Cooperative Oncology Group (ECOG) performance status 0 or 1. However, patients in clinical trials may differ significantly from real-world patients. In the GIDEON (Global Investigation of therapeutic DEcisions in hepatocellular carcinoma and Of its treatment with sorafeNib) study [74], a real-world observational study of sorafenib in advanced HCC, 61% of patients had Child–Pugh Class A (score 5 or 6) and 25% of patients had Child–Pugh B (score 7, 8 or 9). Though the overall type and incidence of adverse events were not significantly different between Child–Pugh Class A and B patients, the incidence of adverse events requiring sorafenib dose reductions was higher in patients with Child–Pugh Class B (40% vs. 29% in Child–Pugh Class A) indicating worse tolerance and shorter duration on therapy. Additionally, the OS was significantly shorter with more advanced Child–Pugh Class. In addition, real-world data revealed that the benefit of sorafenib in old patients (Medicare beneficiaries) with unselected advanced HCC (regardless of Child–Pugh Class) was very minimal, if present at all [75]. Similarly, in a meta-analysis, McNamara et al. [76] commented that sorafenib use is unlikely clinically meaningful for the Child–Pugh Class B status population; the relevant multicenter randomized controlled trial (B Child HCC patients-Optimization Of Sorafenib Treatments (BOOST) trial) was terminated early due to lack of enrollment. 

The overall QoL effect of sorafenib in patients with HCC based on the GIDEON [74], SHARP [7], and the Sorafenib Asia-Pacific trial [8] results is in contrast to the experience so far with sorafenib in renal cell carcinoma, where it showed clinical benefits with symptom improvement without adversely impacting overall QoL when compared with the placebo [77]. Similarly, regorafenib improved survival without significantly worsening QoL in refractory gastric cancer [78] or colorectal cancer [79]. These studies indicate the significant impact of underlying liver disease in terms of drug tolerance.

Third, limiting the use of systemic agents to only patients whom systemic therapies are predicted to benefit and in whom ideal supportive care (with or without early use of hospice) are not expected to derive any clinical benefit will assist in the overall improvement in QoL of patients with advanced HCC. Until now, with the exception of high serum AFP (>400 ng/mL) for ramucirumab, no predictive biomarkers have been identified that can guide the selection of specific systemic agents in clinical practices, although preliminary data indicate that somatic genomic alterations may aid treatment selection, for example, oncogenic PI3K–mTOR pathway alterations and an activated WNT/β-catenin signaling pathway may be negative predictive biomarkers for sorafenib or ICIs treatment, respectively [80]. Additionally, Teufel et al. [81] identified certain plasma protein patterns (i.e., decreased angiopoietin 1 or cystatin B) and miRNAs (i.e., MIR30A, MIR122) that were associated with increased OS following regorafenib treatment in the RESORCE trial. 

Bruix et al. [82] pooled the data from the two sorafenib clinical trials [7,8] and found that sorafenib provided greater survival benefit in patients with disease confined to the liver, HCV infection, and lower neutrophil-to-lymphocyte ratio. Prognostic factors for poorer OS were also identified including ECOG Performance Status 1 or 2 (vs. 0), Barcelona Clinic Liver Cancer Stage C (vs. B), presence of microvascular invasion, high tumor burden, maximum baseline target lesion size >6 cm, AFP >200 ng/mL, high bilirubin, low albumin, albumin-bilirubin grade 2 (vs. 1), high neutrophil-lymphocyte ratio, and low ALP. Further, early decrease of >20% in AFP levels following sorafenib treatment, occurrence of HFSR [83], hypertension, and diarrhea [84] were associated with better overall benefits from sorafenib. These understandings together with increased treatment options and better toxicity management likely contributed to the improved care in patients with HCC over the recent years [85].

Finally, how good are we in measuring QoL in clinical trials and translating these data in a busy clinical practice? Placebo controlled design and use of standardized questionnaires are established methods, but they may not fully capture the big picture. QALY assesses both the quality and quantity of life lived with disease. It is a key indicator for treatment outcomes and an integral part of cost-effectiveness (or cost-utility) analyses of interventions [86]. QALY is the product of multiplying the utility value (health-related quality of life weight) by the years lived in the given state associated with that utility value. For example, 1 QALY means a year of life lived in perfect health. The answer to what is considered cost-effectiveness ratio is debatable though. Azimi and Welch [87] found that researchers usually favor implementing intervention if at the cost of <$61,500 per QALY gained, were against implementing intervention if >$166,000 per QALY gained, and disagreed about cost effectiveness if the cost was between $61,500 and $166,000 per QALY gained. With the use of these systemic agents, overall life expectancy of advanced HCC ranges from 6.5 to 15 months with a PFS time of 2.8 to 5.5 months (Table 1). Cost effective analyses in the available literature revealed seemingly marginal benefits of these medications (Table 2). New cost-effective pharmaceutical agents are needed to improve the care of patients with advanced HCC.

The accelerated approval of nivolumab and pembrolizumab was meant to meet such demand. In addition to promising results in terms of antineoplastic efficacy, nivolumab seemed to halt the deterioration of QoL in advanced HCC while no QoL information was reported in the pembrolizumab trial. Further studies are needed for their use as second-line agents for advanced HCC. Additional research is also warranted to identify positive and negative predictive biomarkers for established agents in advanced HCC. 

Ideal cancer treatment, including best supportive care for disease and treatment-related symptoms, should not only prolong OS but also improve QoL. Side effects of anticancer agents may impact QoL and also cause treatment interruptions and/or dose reductions and thus decrease the optimal efficacy of these medications. This is especially true in the treatment of advanced HCC with sorafenib and regorafenib, which resulted in no better QoL measures. Admittedly, this review may be influenced by our personal viewpoints and literature search strategies; the main limitation is really either the absence of QoL data or/and the inconsistent reporting that does not allow easy cross trial comparisons. Further research is needed to identify cost-effective pharmaceutical agents, optimal protocol of interventions (including predictive biomarkers) and management of drug toxicity and cirrhosis in meeting these goals. Additional research based on real-world data, like the GIDEON study, is also warranted. At the moment, treatment of advanced HCC should continue the current multidisciplinary tumor board format; additionally, it should also involve ancillary specialties and treatments to decrease medication side effects, improve QoL, and tailor the management to meet patients’ need.

## 8. Conclusions

Quality of life in patients with HCC is significantly compromised as compared to the general population. Over the past decade, tremendous development has been made in drug development for advanced HCC. At the moment, seven systemic agents are approved for standard of care use in patients with advanced HCC with preserved performance status and liver function. Based on the available evidence, lenvatinib and ramucirumab may have positive impacts in patients’ QoL whereas for other systemic treatments, no evidence supports a positive benefit on QoL. ICIs have promising activity in the second-line setting with a manageable toxicity profile and preservation of QoL. 

All studies in the approved agents showed survival benefit in a highly selected population with Child–Pugh Class A and ECOG performance status 0 or 1. However, patients with a more advanced Child–Pugh Class have worse outcomes in adverse events and OS. These findings could be related to drug toxicity and/or ongoing cirrhotic liver impairment with or without tumor stabilization. QoL measures should be included in all clinical trials and taken into consideration when selecting systemic treatments for advanced HCC. The management of patients with advanced HCC should not only include treatments targeting HCC, but also the management of treatment toxicity and other supportive care.

## Figures and Tables

**Table 1 cancers-11-00861-t001:** A summary of outcomes, side effects, and quality of life in systematic treatments of hepatocellular carcinoma.

Systemic Treatment	Efficacy: OS (months), PFS (months), %ORR. Treatment vs. Control	Incidence of Common Drug-Related Adverse Events *	Grade 3 and 4 Drug-Related Toxicities	Quality of Life Assessment	Ref.
Sorafenib (vs. placebo)	OS: 10.7 vs. 7.9 (*p* < 0.001) PFS: radiographic (RECIST) = 5.5 vs. 2.8 (*p* < 0.001); symptomatic = 4.1 vs. 4.9 (*p* = 0.77) ORR: 2% vs. 1% (SD = 71% vs. 67%)	Diarrhea (39%), fatigue (22%), HFSR (21%), rash or desquamation (16%), alopecia (16%), anorexia (14%), nausea (16%), weight loss (9%), abdominal pain (8%), dry skin (8%), pruritus (8%), bleeding (7), voice changes (6%), hypertension (5%), vomiting (5%)	HFSR (8%), hypertension (2%), abdominal pain (2%), and weight loss (2%)	There was a lack of a significant difference in responses to the FHSI8 questionnaire.	[7]
Sorafenib (vs. placebo)	OS: 6.5 vs. 4.2 (*p* = 0.014) PFS: radiographic (RECIST) = 2.8 vs. 1.4 (*p* = 0.0005); symptomatic = 3.5 vs. 3.4 months (*p* = 0.50) ORR: 3.3% vs. 1.3% (SD = 54% vs. 27.6%)	HFSR (45%), diarrhea (25.5%), alopecia (24.8%), fatigue (20.1%), rash or desquamation (20.1%), hypertension (18.8%), anorexia (12.8%), and nausea (11.4%)	HFSR (10.7%), diarrhea (6.0%), and fatigue (3.4%)	Both groups had similar total scores on the FSHI-8 questionnaire and scores with the FACT–Hep questionnaire showed no difference in quality of life between the two groups.	[8]
Lenvatinib (vs. sorafenib)	OS: 13.6 vs. 12.3 (HR 0.92, 95% CI 0.79–1.06) PFS: radiographic (mRECIST) = 7.4 vs. 3.7 (*p* < 0.0001); time to progression = 7.9 vs. 3.7 (*p* < 0.0001) ORR: 24.1% vs. 9.2% (SD = 51% vs. 51%)	Lenvatinib: hypertension (42%), diarrhea (39%), decreased appetite (34%), decreased weight (31%), fatigue (30%), HFSR (27%), and proteinuria (25%).Sorafenib: HFSR (52%), diarrhea (46%), hypertension (30%), decreased appetite (27%), alopecia (25%), and fatigue (25%).	Lenvatinib: hypertension (23%), decreased weight (8%), increased blood bilirubin (7%), proteinuria (6), decreased platelet (5%), and decreased appetite (5%). Sorafenib: hypertension (14%), HFSR (11%), elevated AST (8%), increased blood bilirubin (5%)	The time to clinically meaningful deterioration in role functioning, pain, diarrhea, and nutrition was later with lenvatinib than sorafenib, but the summary score of EORTC QLQ-C30 was not significantly different between the two treatments (*p* = 0.07).	[9]
Regorafenib (vs. placebo)	OS: 10.6 vs. 7.8 (*p* < 0.0001) PFS: Radiographic (mRECIST) = 3.1 vs. 1.5; time to progression = 3.2 vs. 1.5 ORR = 11% vs. 4% (SD = 54% vs. 32%)	HFSR (52%), diarrhea (33%), fatigue (29%), anorexia (24%), hypertension (23%), increased blood bilirubin (19%), increased AST (13%), oral mucositis (11%), nausea (11%), abdominal pain (9%), hoarseness (9%)	Regorafenib: Hypertension (13%), HFSR (13%), increased blood bilirubin (7%), fatigue (6%), increased AST (5%), and hypophosphatemia (5%) Placebo: increased AST (5%), hypertension (3%), increased blood bilirubin (2%), fatigue (2%), HFSR (1%)	No difference was found between interventions for quality of life measures with the exception of the FACT-Hep total score favoring placebo.	[10]
Cabozantinib (vs. placebo)	OS: 10.2 vs. 8.0 (*p* = 0.005) PFS: radiographic (RECIST) = 5.2 vs. 1.9 (*p* < 0.001); time to progression = not reported ORR = 4% vs. <1% (SD = 60% vs. 33%)	# Diarrhea (54%), decreased appetite (48%), HFSR (46%), fatigue (45%), nausea (31%), hypertension (29%), vomiting (26%), increased AST (22%), asthenia (22%), dysphonia (19%), constipation (19%), abdominal pain (18%), weight loss (17%), increased ALT (17%), mucositis (14), pyrexia (14), abdominal pain (13%), cough (13%), peripheral edema (13%), stomatitis (13%)	#Cabozantinib: HFSR (17%), hypertension (16%), increased AST level (12%), fatigue (10%), diarrhea (10%), and decreased appetite (6%). Placebo: increased AST level (7%), ascites (5%), abdominal pain (4%), fatigue (4%)	Not reported	[11]
Ramucirumab (vs. placebo)	OS: 8.5 vs. 7.3 (*p* = 0.0199) PFS: radiographic (RECIST) = 2.8 vs. 1.6 (*p* < 0.0001); time to progression = 3.0 vs. 1.6 (*p* < 0·0001) ORR = 4.6% vs. 1.1% (SD = 55.3% vs. 37.9%)	Hypertension (17%), proteinuria (14%), fatigue (14%), nausea (12%), bleeding events (11%), decreased appetite (11%), liver injury or failure (8%), peripheral edema (8%)	Ramucirumab: hypertension (8%), liver injury or failure (4%), proteinuria (2%), fatigue (1%), and peripheral edema (1%) Placebo: hypertension (2%), diarrhea (1%)	Median time to deterioration in FHSI-8 total scores (3.7 months vs. 2.8 months (*p* = 0.238) and ECOG performance statuses (*p* = 0.77) did not differ between groups.	[12]
Nivolumab (single arm)	OS: 15.0 (dose escalation); 13.2 (dose expansion) PFS: radiographic 3.4 (dose escalation); 4.1 (dose expansion) ORR: 15% (dose escalation); 20% (dose expansion)	Most commonly included fatigue, diarrhea, pruritus, rash and liver function test abnormalities. Serious side effects of pemphigoid, adrenal insufficiency, and liver disorder were also reported		No significant change from baseline.	[18]
Pembrolizumab (single arm)	OS: 12.9 PFS: radiographic (irRECIST) = 4.9 ORR: 17%	Common treatment-related side effects included fatigue (21%), increased AST (13%), pruritus (12%), diarrhea (11%), and rash (10%). Grade 3 events included fatigue (4%), increased AST (7%), and increased ALT (4%). One grade 4 occurrence of hyperbilirubinemia was also reported (not related to treatment).		Not reported	[19]
Pembrolizumab (vs. placebo)	OS: 13.9 vs. 10.6 (*p* = 0.0238) ^@^ PFS: radiographic (RECIST) = 3 vs. 2.8 (*p* = 0.0186) ORR: 18.3% vs. 4.4%	Incidence of hepatitis and other immune mediated events were generally consistent with those previously reported in studies.		Not reported	[26]

Note: OS = overall survival; ORR = overall response rate; HR = hazard ratio; PFS= progression free survival; SD = stable disease; HCC = hepatocellular carcinoma; ECOG = Eastern Cooperative Oncology Group; FHSI-8 = Functional Hepatobiliary Symptom Index-8; FACT-Hep = Functional Assessment of Cancer Therapy-Hepatobiliary questionnaire; EORTC QLQ-C30 = European Organization for Research and Treatment of Cancer Quality of Life Questionnaire—C30; HFSR = hand-foot skin reaction; AST = aspartate aminotransferase; ALT = alanine aminotransferase. * No information of the placebo group was provided due to space limitations. # Adverse events regardless of causality. ^@^ The difference did not meet significance per the prespecified statistical plan.

**Table 2 cancers-11-00861-t002:** Cost effectiveness data of systematic treatments. *

Systemic Treatment	Number of Quality-Adjusted Life Year	Costs per Quality-Adjusted Life Year (USD)	Reference
Sorafenib	0.18 to 0.53	–$1,014,507 to $934,803	[27]
Lenvatinib	0.23	$45,477.25	[28]
Regorafenib	0.18 to 0.25	$201,797 to $268,506	[29,30]
Cabozantinib	0.092 through follow-up period	Not available	[31]

* No data available in the literature for ramucirumab, nivolumab, and pembrolizumab in hepatocellular carcinoma.

**Table 3 cancers-11-00861-t003:** Management of drug toxicities and side effects.

Side Effects	Management
Skin rash and HFSR	Callus removal and minimizing friction and direct trauma, keratolytic creams (containing urea, α-hydroxy acids or salicylic acid), topical corticosteroids (clobetasol 0.05% ointment, cortisone cream), analgesics (e.g., lidocaine 2%), and systematic strategies (e.g., pyridoxine 50–150 mg/day to reduce symptoms)
Diarrhea	Loperamide of 4 mg followed by 2 mg every 2 h until 2 h after the last bowel movement, lactulose dose adjustment, opiates (e.g., codeine), and cholestyramine 4 g three times a day
Fatigue	Psychostimulants (methylphenidate and modafinil), treatment for pain, emotional distress, anemia and hypothyroidism, physical exercise, psychoeducation, cognitive behavioral therapy, other mind-body therapies, and activity enhancement
Hypertension	Standard medical practice, non-selective beta blockers (e.g., carvedilol, nadolol or propranolol) followed by calcium channel blockers if noneffective
Pain	Opioids for acute or short-term use, topical therapies, physical therapy, mind-body therapies, anticonvulsants, antidepressants, procedural injections, device implantation, corticosteroids, and radiation therapy
Weight loss and malnutrition	Targeted nutritional supplementation including parenteral nutrition, enteral nutrition or oral nutritional supplements (not appropriate for advanced HCC), physical exercise and pharmacological interventions of corticosteroids (dexamethasone, prednisolone, and methylprednisolone), and progesterone analogs (megestrol acetate and medroxyprogesterone acetate)
Severe or refractory adverse events	Dose adjustment or even drug discontinuation may be appropriate, and re-escalation to full dose when possible is recommended
Immune-mediated toxicities in immune checkpoint inhibitor therapy	
Grade 2 (moderate)	Temporarily withhold and resume treatment if symptoms or toxicity decrease, prednisone 0.5 mg/kg/day or equivalent if symptoms do not resolve within a week
Grade 3 or 4 (severe or life-threatening)	Permanently discontinue treatment, prednisone 1 to 2 mg/kg/day or equivalent with a taper over at least one month after symptom relief
Liver function abnormalities	Withhold for AST or ALT >2.5 times the upper limit of normal (ULN) but ≤5 times the ULN, or total bilirubin >1.5 times the ULN but ≤3 times the ULN; permanently discontinue for AST or ALT >5 times the ULN, or total bilirubin >3 times the ULN

Note: HFSR = hand-foot skin reaction; Grade 2 (moderate) = immune-mediated toxicities affecting quality of life and requiring intervention; Grade 3 or 4 (severe or life-threatening) = severe or life-threatening immune-mediated toxicities; ULN = upper limit normal.

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
