# Peer review of "Outcomes and Quality of Life of Systemic Therapy in Advanced Hepatocellular Carcinoma"

_cancers, 2019, doi:10.3390/cancers11060861_

Round 1
Reviewer 1 Report
In the current review the authors discuss the outcome and quality of life of HCC patietns treated with systemic therapy, along with the challenge of underlying liver cirrhosis which impacts on prognosis and management.
General comments:
Overall, the paper lacks some depth. The authors simply summarize the available data for HCC superficially and in a way that it appears to be just a listing of references. A clear story line and interpretation of the data is missing. This makes the first part of the paper hard to read; this part certainly needs some attention. The second part on toxicity management is better.
Specific comments:
In the abstract, the authors should also acknowledge the positive results of the REACH-2 trial showing an improvement in OS over placebo in second-line.
The authors should check toxicity results in table 1. For example, for sorafenib (ref. 7), side effects do not seem to be correctly reported: HFS occurred in 21%, diarrhea in 39%, weight loss in 9% etc. (Note: in this trial, side effects in the original NEJM publication are given as percentages, not as numbers!). For cabozantinib, they only report percentages for grade 3-5 adverse events while any grade adverse events are reported in for other agents (e.g., sorafenib). Report either any grade OR high grade adverse events for the different drugs in your table 1, but be consistent. Anyway, I highly recommend to check the numbers in this table again.
The authors should discuss the KEYNOTE-240 phase III study, which tested pembrolizumab vs placebo in second-line. According to a press release, the study failed on its co-primary endpoint OS and PFS. The results will be presented at ASCO 2019, the abstract is already available via ASCO website.
The chapter ‘The role of cirrhosis’ is very superficial and not very helpful for clinicians. The authors should include current recommendations (e.g., latest BAVENO consensus) for the management of liver cirrhosis, portal hypertension and their complications.
The authors should discuss the role of side effects (e.g., dermatological side effects) as predictors of outcome. Several studies reported improved survival with sorafenib in those who experienced hand-foot-skin reaction. These data were recently summarized in a meta-analysis (Aliment Pharcmacol Ther 2019;49:482).
The authors should also discuss the impact of dose reduction of sorafenib for on OS (see Iavarone et al. Hepatology 2011;54:2055).
When discussing management of hypertension, the authors may also consider discussing a potentially beneficial role of renin-angiotensin-system inhibitors in different solid tumors (Sci Transl Med 2017;9:eaan5616), including HCC (United European Gastroenterol J 2017;5:987).
Some typos/grammar errors need to be corrected. I recommend to thoroughly screen the paper for typos again.
Author Response
Comments and Suggestions for Authors
In the current review the authors discuss the outcome and quality of life of HCC patients treated with systemic therapy, along with the challenge of underlying liver cirrhosis which impacts on prognosis and management.
General comments:
Overall, the paper lacks some depth. The authors simply summarize the available data for HCC superficially and in a way that it appears to be just a listing of references. A clear story line and interpretation of the data is missing. This makes the first part of the paper hard to read; this part certainly needs some attention. The second part on toxicity management is better.
Answer: This review aimed to review the available evidence of quality of life with current systemic therapy options for advanced HCC. As such, in the limited space available (6,000 words), we mainly focused on reviewing quality of life and side effects of these treatments in the FDA approved agents, and the treatment of cirrhosis rather going into details on the development and efficacy of these agents. We have expanded the sections where QoL data for each agent were presented.
In the revised manuscript, we added possible sorafenib prognostic factors for better outcomes, though prognostic and not predictive, as it may assist help understand who has a better chance of benefiting from treatment and thus a more favorable risk/benefit ratio. We also reported recent data on molecular biomarkers, promising as potential future negative predictive biomarkers (ref 70). At this time only AFP >400 is a relevant biomarker in HCC allowing use of ramucirumab in second line. We also expanded the section on treatment of common complications of cirrhosis.
Specific comments:
In the abstract, the authors should also acknowledge the positive results of the REACH-2 trial showing an improvement in OS over placebo in second-line.
Answer: In the revised manuscript, we added “and ramucirumab”.
The authors should check toxicity results in table 1. For example, for sorafenib (ref. 7), side effects do not seem to be correctly reported: HFS occurred in 21%, diarrhea in 39%, weight loss in 9% etc. (Note: in this trial, side effects in the original NEJM publication are given as percentages, not as numbers!). For cabozantinib, they only report percentages for grade 3-5 adverse events while any grade adverse events are reported in for other agents (e.g., sorafenib). Report either any grade OR high grade adverse events for the different drugs in your table 1, but be consistent. Anyway, I highly recommend to check the numbers in this table again.
Answer: In the revised manuscript, we went over the published studies including their appendices and updated the data. We also reorganized the table to make it more reader friendly.
The authors should discuss the KEYNOTE-240 phase III study, which tested pembrolizumab vs placebo in second-line. According to a press release, the study failed on its co-primary endpoint OS and PFS. The results will be presented at ASCO 2019, the abstract is already available via ASCO website.
Answer: In the revised manuscript, we added a new paragraph depicting the findings of the KEYNOTE-240 trial as the following: “The preliminary results of the phase III study (KeyNote-240) of pembrolizumab versus best supportive care for second line therapy in advanced HCC were presented in the 2019 ASCO Annual Meeting [26]; pembrolizumab improved OS (HR: 0.78; one sided p = 0.0238) and PFS (HR: 0.78; one sided p = 0.0209) with a manageable toxicity profile compatible with prior experience with ICIs in advanced HCC; however, these differences did not meet significance per the prespecified statistical plan. QoL information was reported in neither Keynote-224 nor Keynote-240.” We also added this information in Table 1.
The chapter ‘The role of cirrhosis’ is very superficial and not very helpful for clinicians. The authors should include current recommendations (e.g., latest BAVENO consensus) for the management of liver cirrhosis, portal hypertension and their complications.
Answer: We reviewed the EASL Clinical Practice Guidelines and added (revised) the relevant paragraphs as “Treatment of compensated cirrhosis usually aims at the underlying liver disease including treatment of viral hepatitis, alcohol abstinence, and avoidance of hepatotoxic drugs in addition to symptom control. In decompensated cirrhosis, the removal of etiological factors, particularly alcohol consumption and hepatitis B or C virus infection, is associated with decreased risk of decompensation and increased OS [47]. It requires additional complication-specific interventions, for example, nonselective beta blocker and endoscopic band ligation for varices; vasoactive drugs (terlipressin, somatostatin or octreotide) together with endoscopic variceal ligation or sclerotherapy for variceal bleeding; diuretics (spironolactone alone or combination with furosemide or torsemide) and sodium restriction (80–120 mmol/day, corresponding to 4.6–6.9 g of salt) for ascites; terlipressin in combination with albumin (non-intensive care unit) or norepinephrine (intensive care unit) and less favorably midodrine, octreotide, and albumin for hepatorenal syndrome; antibiotic treatment and prophylaxis for spontaneous bacterial peritonitis; albumin or baclofen administration (10 to 30mg/day) for muscle cramps; and transjugular intrahepatic portosystemic shunts (TIPS) for portal hypertension and refractory variceal bleeding [47]. TIPS is associated with the occurrence of shunt stenosis and hepatic encephalopathy. The management of hepatic encephalopathy targets plasma ammonia primarily by reducing its production and absorption. Lactulose and rifaximin have remained the mainstay of treatment [48]. Additionally, repeated large volume paracentesis plus albumin (8 g/L of ascites removed) may be required for refractory ascites, and the use of nonselective beta blockers should be cautious in severe or refractory ascites [47].”
“…Nonetheless, in a recent study, Cabibbo et al [56] argues that treatment of hepatitis C in patients with "active" HCC remains controversial as these patients were usually excluded in clinical trials. For HBV-related HCCs, treatment of hepatitis can preserve or improve liver function, and may be beneficial for patients undergoing curative therapy, or locoregional therapy/chemotherapy with reasonable life expectancy [57].”
The authors should discuss the role of side effects (e.g., dermatological side effects) as predictors of outcome. Several studies reported improved survival with sorafenib in those who experienced hand-foot-skin reaction. These data were recently summarized in a meta-analysis (Aliment Pharcmacol Ther 2019;49:482). The authors should also discuss the impact of dose reduction of sorafenib for on OS (see Iavarone et al. Hepatology 2011;54:2055).
Answer: We reorganized the Discussion section with the above reviewer mentioned information included as the following “Third, limiting the use of systemic agents to patients predicted to benefit the most and use of ideal supportive care only (with or without early use of hospice) in patients not expected to derive any clinical benefit will assist in overall improvement in QoL of patients with advanced HCC. Until now, with the exception of high serum AFP (>400 ng/ml) for ramucirumab, no predictive biomarkers have been identified that can guide the selection of specific systemic agents in clinical practices, although preliminary data indicate that somatic genomic alterations may aid treatment selection, for example, oncogenic PI3K–mTOR pathway alterations and an activated WNT/β-catenin signaling pathway may be negative predictive biomarkers for sorafenib or ICIs treatment, respectively [78]. Additionally, Additionally, Teufel et [79] al identified certain plasma protein patterns (i.e., decreased angiopoietin 1 or cystatin B) and miRNAs (i.e., MIR30A, MIR122) associated with increased OS following regorafenib treatment in the RESORCE trial. Bruix et al [78] pooled the data from the two sorafenib clinical trials [7, 8] and found that sorafenib provided greater survival benefit in patients with disease confined to the liver, HCV infection, and lower neutrophil-to-lymphocyte ratio. Prognostic factors for poorer OS were also identified including ECOG Performance Status 1 or 2 (vs. 0), Barcelona Clinic Liver Cancer Stage C (vs. B), presence of microvascular invasion, high tumor burden, maximum baseline target lesion size >=6 cm, AFP >200 ng/ml, high bilirubin, low albumin, albumin-bilirubin grade 2 (vs. 1), high neutrophil-lymphocyte ratio, and low ALP. Further, early decrease of > 20% in AFP levels following sorafenib treatment, occurrence of HFSR [80], hypertension and diarrhea [81] have been associated with better overall benefits from sorafenib. These understandings together with the increased treatment options and better toxicity management likely contributed to the improved care in patients with HCC treated in a single center over the recent years [82].
When discussing management of hypertension, the authors may also consider discussing a potentially beneficial role of renin-angiotensin-system inhibitors in different solid tumors (Sci Transl Med 2017;9:eaan5616), including HCC (United European Gastroenterol J 2017;5:987).
Answer: In the revised manuscript, we added “Interestingly, based on preclinical studies and small controlled trials, renin-angiotensin system inhibitors may have antitumor effects (likely related to VEGF response) and improve both liver fibrosis and portal hypertension; their use correlated with a gain of 5 months' in median OS in patients with advanced HCC [59].”
Some typos/grammar errors need to be corrected. I recommend to thoroughly screen the paper for typos again.
Answer: We went through the revised manuscript and corrected them.
Reviewer 2 Report
I learned a lot from your excellent manuscript. I have some comments and questions. The reader will be able to learn widely about the outcome and the QOL of the patient treated by MTKI.
#1 What is the definition of Quality of life in this review? Please check and add.
#2 L72 Method: Why did you identify phase III trials of MTKI? One of your focus of this review might be Quality of Life of HCC patients. Phase I, II, or III trials might include the findings of QOL, however, it will not be the main focus of these trials. Please add the significance of this protocol to search for QOL.
#3 At the result part, you mentioned OS, PFS, ORR, and side effects as the outcomes. Some studies did not include QOL assessments. I wonder what was your focus of this study. If the focus of this study is the outcomes of phase I, II, and III trials, I wonder what is the originality of this study compared with previous medical studies.
#4 At the discussion part, I could not detect the limitations of this study. Please check and revise.
Author Response
I learned a lot from your excellent manuscript. I have some comments and questions. The reader will be able to learn widely about the outcome and the QOL of the patient treated by MTKI.
#1 What is the definition of Quality of life in this review? Please check and add.
Answer: thanks for the kind suggestions. We added the definition of quality of life per WHO 1947 definition as “QoL refers to the state of complete physical, mental, and social wellbeing, and not merely the absence of disease or infirmity.”
#2 L72 Method: Why did you identify phase III trials of MTKI? One of your focus of this review might be Quality of Life of HCC patients. Phase I, II, or III trials might include the findings of QOL, however, it will not be the main focus of these trials. Please add the significance of this protocol to search for QOL.
Answer: Our review focuses on systemic treatments of HCC. Our search for systemic medications was guided by the FDA approval information. Admittedly, phase I, II, and III trials may all include QoL findings; however, we only included phase III trials for MTKIs with positive primary outcomes and phase II/III trials for immunotherapy as the sample size is larger and that data more readily available compared to earlier phase development studies where they are not routinely collected. We revised the Methods section as the following: “Using the search logic ("quality of life") AND hepatocellular carcinoma [Title/Abstract], we identified 667 articles in PubMed. Abstracts were then screened to identify outcome and QoL results. The search for systemic medications was guided by the FDA approval information [13-17][20][21]. Specifically, we searched for and identified all phase III trials in PubMed for targeted therapy for HCC with proven OS benefit (including sorafenib, cabozantinib, lenvatinib, regorafenib, and ramucirumab [7-12]). Using the search logic ((immune therapy) OR immunotherapy) AND hepatocellular carcinoma [Title] with article type limited to clinical trials, we were able to locate 80 articles with immune checkpoint inhibitors (ICIs) in PubMed, and we were only able to identify and include two single-arm phase II trials [18, 19]. Finally, we searched Google and Google Scholar for the same information; we were able to identify and include one additional phase III trial testing an FDA approved agent (KeyNote-240) [26]. Search was carried out from inception to June 6, 2019. Data on outcomes, QoL, and toxicities in advanced HCC of these trials were extracted in Table 1. Additionally, we also searched for cost effectiveness analyses of these medications and relevant information, if any, was described and presented separately in Table 2.”
#3 At the result part, you mentioned OS, PFS, ORR, and side effects as the outcomes. Some studies did not include QOL assessments. I wonder what was your focus of this study. If the focus of this study is the outcomes of phase I, II, and III trials, I wonder what is the originality of this study compared with previous medical studies.
Answer: Our review focuses in quality of life with approved systemic agents with advanced HCC and side effect/cirrhosis management. The main focus of the review is not OS or PFS. We briefly discuss efficacy data as it is important for understanding the risks and benefits with use of these agents as we do routinely in clinic with our patients. The tradeoff between risks and benefits can be significant and clinical trial data cannot always be translated 100% into clinic practices where sicker patients are treated with the same agents. As compared to previous reviews, this review is most up-to-date with an emphasis on adverse event management and quality of life.
#4 At the discussion part, I could not detect the limitations of this study. Please check and revise
Answer: Thank you for the comment. In the revised manuscript, we added “Admittedly, this review may be influenced by our personal viewpoints and literature search strategies; the main limitation of this review is really either the absence of QoL data or/and the inconsistent reporting that does not allow easy cross trial comparisons in the published trials.”
Reviewer 3 Report
I read with pleasure this review regarding QoL in patients with hepatocellular carcinoma treated with systemic treatments. The concept behind the paper is original and the information provided by the Authors are helpful in a clinical scenario which now includes many different FDA-approved drugs. I have only a few comments:
1) Introduction, line 52: " Although these new agents increased the overall survival (OS)...". In its current place, this sentence seems to imply that also nivolumab and pembrolizumab increased the OS, which is not properly correct (in this purpose, the Phase 3 pembrolizumab trial failed this endpoint). Please modify the structure of this paragraph.
2) Introduction, line 55: " At the same time, even with the developing technology and advancement in cancer pharmacotherapy, the 5 year survival rate of HCC in the United States is only 17.7%, 2 years survival rate is less than 50% and 5-year survival is only 10% with a median survival following diagnosis of approximately 6 to 20 months". This sentence is confusing and contradictory (the 5-year survival rate is 17.7 or 10%?). Please check the data and correct this sentence.
3) Table 1 is very hard to digest and should be simplified.
4) Pain management, lines 249-256. The problem of opioid use in cirrhotic patients is very complex. On one hand, cirrhotic patients are more prone to receiving opioid prescriptions (doi: 10.1136/bmjgast-2018-000271). On the other hand, pain in cirrhosis is still undertreated as many physician still fear the risk of hepatic encephalopathy (doi: 10.5812/hepatmon.23539). The Authors can expand this session.
5) Pain management, lines 249-256. The Authors state that morphine and fentanyl are good choices for patients with liver function impairment. There are however initial reports suggesting a good safety and efficacy of the combination oxycodone/naloxone that the Authors could discuss (doi: 10.1111/liv.13546)
6) Discussion: the Authors correctly focus on the role of proper management of toxicities in maintaining the quality of life. There are initial reports about the impact of operators experience in the management of toxicities that could be considered to augment the discussion (doi: 10.1159/000497161)
7) Discussion: There are currently 5 different drugs with permanent FDA approval for hepatocellular carcinoma (sorafenib, lenvatinib, regorafenib, cabozantinib, lenvatinib). Until now, no biomarkers predictive of response have been identified. So, in most cases, there are no clear drivers which can help the physician toward one specific drug or another (for example lenvatinib vs sorafenib or regorafenib vs cabozantinib). Do the Authors think that more QoL-focused studies might help the physicians in their clinical practice?
Author Response
I read with pleasure this review regarding QoL in patients with hepatocellular carcinoma treated with systemic treatments. The concept behind the paper is original and the information provided by the Authors are helpful in a clinical scenario which now includes many different FDA-approved drugs. I have only a few comments:
1) Introduction, line 52: " Although these new agents increased the overall survival (OS)...". In its current place, this sentence seems to imply that also nivolumab and pembrolizumab increased the OS, which is not properly correct (in this purpose, the Phase 3 pembrolizumab trial failed this endpoint). Please modify the structure of this paragraph.
Answer: Pembrolizumab appears to benefit patients in the second line setting with limited toxicity based on the recently reported KEYNOTE 240 phase III study. The OS benefit was not statistically significant (based on a very stringent p value) but appears clinically meaningful. Nonetheless, we modified the relevant sentence as “While the overall survival (OS) for patients with advanced HCC has increased, these treatments are not curative [3] and the unique treatment-related toxicities can further compromise patients’ delicate health.”
2) Introduction, line 55: " At the same time, even with the developing technology and advancement in cancer pharmacotherapy, the 5 year survival rate of HCC in the United States is only 17.7%, 2 years survival rate is less than 50% and 5-year survival is only 10% with a median survival following diagnosis of approximately 6 to 20 months". This sentence is confusing and contradictory (the 5-year survival rate is 17.7 or 10%?). Please check the data and correct this sentence.
Answer: Due to word limit requirements, we deleted all the information you mentioned.
3) Table 1 is very hard to digest and should be simplified.
Answer: thanks for the comments, we tried to simplify Table 1 in the revised manuscript.
3) Pain management, lines 249-256. The problem of opioid use in cirrhotic patients is very complex. On one hand, cirrhotic patients are more prone to receiving opioid prescriptions (doi: 10.1136/bmjgast-2018-000271). On the other hand, pain in cirrhosis is still undertreated as many physician still fear the risk of hepatic encephalopathy (doi: 10.5812/hepatmon.23539). The Authors can expand this session.
Answer: Thank you for the kind suggestions. Original text in the manuscript “Pain management in patients with advanced HCC could be more liberating in opioid use (as compared to chronic non-cancer pain) [61].” We added "Meanwhile, pain in these patients remains undertreated for fears of an increased risk of toxicity."
Pain management, lines 249-256. The Authors state that morphine and fentanyl are good choices for patients with liver function impairment. There are however initial reports suggesting a good safety and efficacy of the combination oxycodone/naloxone that the Authors could discuss (doi: 10.1111/liv.13546)
Answer: We added "....and oxycodone/naloxone can be a safe and effective alternative" in the relevant section
6) Discussion: the Authors correctly focus on the role of proper management of toxicities in maintaining the quality of life. There are initial reports about the impact of operators experience in the management of toxicities that could be considered to augment the discussion (doi: 10.1159/000497161)
Answer: We added " These understandings together with the increased treatment options and better toxicity management likely contributed to the improved care in patients with HCC the recent years.”
7) Discussion: There are currently 5 different drugs with permanent FDA approval for hepatocellular carcinoma (sorafenib, lenvatinib, regorafenib, cabozantinib, lenvatinib). Until now, no biomarkers predictive of response have been identified. So, in most cases, there are no clear drivers which can help the physician toward one specific drug or another (for example lenvatinib vs sorafenib or regorafenib vs cabozantinib). Do the Authors think that more QoL-focused studies might help the physicians in their clinical practice?
Answer: Surely studies like GIDEON will assist by collecting real world data. A similar observational study for regorafenib is in our knowledge underway (https://clinicaltrials.gov/ct2/show/NCT03289273). More importantly, more studies are urgently needed to identify relevant predictive rather than prognostic biomarkers. We added "Until now, with the exception of high serum AFP (>400 ng/ml) for ramucirumab, no predictive biomarkers have been identified that can guide the selection of specific systemic agents in clinical practices, although preliminary data indicate that somatic genomic alterations may aid treatment selection, for example, oncogenic PI3K–mTOR pathway alterations and an activated WNT/β-catenin signaling pathway may be negative predictive biomarkers for sorafenib or ICIs treatment, respectively [77]. Additionally, Teufel et al [78] identified certain plasma protein patterns (i.e., decreased angiopoietin 1 or cystatin B) and miRNAs (i.e., MIR30A, MIR122) were associated with increased OS following regorafenib treatment in the RESORCE trial.”
At the moment, we think we should take into consideration of QoL when selecting systematic agents. In the discussion section, we commented “Further research is needed to identify cost-effective pharmaceutical agents, optimal protocol of interventions (including predictive biomarkers) and management of drug toxicity and cirrhosis in meeting these goals. Additional research based on real world data, like the GIDEON study, is also warranted. At the moment, treatment of advanced HCC should continue the current multidisciplinary tumor board format; additionally, it should also involve ancillary specialties and treatments to decrease medication side effects, improve QoL, and tailor the management to meet patients’ need.”
Round 2
Reviewer 1 Report
The authors revised the paper, which improved the quality of the work, however I have still 2 more points that should be addressed:
Table 1: As I said in my first round of comments, numbers in table 1 for sorafenib ref 7 are WRONG!!!. HFS occurred in 21%, diarrhea in 39%, weight loss in 9% etc. (Note: in this trial, side effects in the original NEJM publication are given as percentages, not as numbers!). I haven’t checked the other studies listed in this table for errors, this is what the authors have to do!
The paragraph on cirrhosis hasn’t really improved. Now, you just list the recommendations for management of cirrhosis without putting it into context, it is hard to read. I would also start the chapter with an introduction on why liver cirrhosis is important for HCC patients, how man HCC patients suffer from cirrhosis, etc. You may wanna check out the review paper on cancer and cirrhosis again that may help to improve this chapter: ESMO open 2016;1:e000042. Otherwise, delete the section on cirrhosis completely.
Author Response
The authors revised the paper, which improved the quality of the work, however I have still 2 more points that should be addressed:
Table 1: As I said in my first round of comments, numbers in table 1 for sorafenib ref 7 are WRONG!!!. HFS occurred in 21%, diarrhea in 39%, weight loss in 9% etc. (Note: in this trial, side effects in the original NEJM publication are given as percentages, not as numbers!). I haven’t checked the other studies listed in this table for errors, this is what the authors have to do!
Answer: thanks for the note, it was our mistake that we mistook the percentages as numbers; we have thus corrected. Additionally, we went through the side effect columns of Table 1 and confirm that the places you mentioned are the only places for such mistake. However, we did add some side effects for the study of Lenvatinib together with deleting "diarrhea (3%)" and changing increased AST from 5% to 6% for regorafenib.
The paragraph on cirrhosis hasn’t really improved. Now, you just list the recommendations for management of cirrhosis without putting it into context, it is hard to read. I would also start the chapter with an introduction on why liver cirrhosis is important for HCC patients, how man HCC patients suffer from cirrhosis, etc. You may wanna check out the review paper on cancer and cirrhosis again that may help to improve this chapter: ESMO open 2016;1:e000042. Otherwise, delete the section on cirrhosis completely.
Answer: thanks for your suggestions; we have made appropriate changes in the paragraph on cirrhosis.
We added “Cirrhosis affects approximately 0.27% of the population, corresponding to 633,323 adults in the United States (based on 2010 US census data). Compared to matched controls, patients with cirrhosis has 18% higher risks of mortality per 2-year interval [47]. Cirrhotic liver has reduced effective blood flow through intrahepatic shunts and sinusoidal capillarization. It decreases CYP enzyme secretions, alters enzyme inductions, and impairs the clearance of drugs [48]. Approximately, 80% of patients with HCC have underlying cirrhosis [4]. Consequently, systemic treatments of HCC will often inevitably involve considerations and treatment of cirrhosis.” The significance of liver dysfunction is also presented in the discussion session with GIDEON study began with the sentence “Second, performance status and baseline liver function can significantly affect outcomes”.